# The Sohanjana Antibullying Intervention: Pilot Results of a Peer-Training Module in Pakistan

## Sohni Siddiqui * and Anja Schultze-Krumbholz

Department of Educational Psychology, Technische Universität Berlin, 10587 Berlin, Germany;
anja.schultze-krumbholz@tu-berlin.de
* Correspondence: s.zahid@campus.tu-berlin.de

**Abstract:** Although comprehensive school-wide interventions targeting bullying have proven effective, many schools, particularly those in low-to-middle income countries like Pakistan, lack the necessary resources to implement them. As a result, implementing cost-effective antibullying bystander programs that train students to become peer advocates is a promising approach for intervening in bullying incidents. Peer training in antibullying intervention involves training students to take on leadership roles and advocate for antibullying initiatives within their school communities. The aim of this study was to design, implement, and evaluate a peer-training module that was subsequently implemented in four schools (N = 38, boys = 26, girls = 12) in Pakistan. The aim was to raise awareness and prepare a team of peer mentors to effectively implement antibullying policies in educational institutions. Pre-post tests and participant feedback were used to evaluate the effectiveness of the peer-training program in increasing knowledge and awareness of bullying and the role of bystanders among trained peer mentors. To determine the impact of the peer mentor training on improving the school environment, data were also collected from students (N = 219, boys = 152, girls = 67) before and after the implementation of the program. The pilot implementation of the peer-training module was considered successful, indicating that the training was effective in improving the knowledge of peer mentors and could be used in a similar way in future cases. The results also showed a significant improvement in the development of prosocial bystanders. However, although improvements were observed in other aspects of the school environment, these did not reach statistical significance. The discussion section includes recommendations and explanations for differences based on gender and urban–rural factors. Suggestions are also made to improve the results for future applications.

**Keywords:** peer antibullying training; bystanders' roles and responsibilities; pilot program; contextualized intervention

## 1. Introduction

Bullying in schools is a major social problem that affects the nation as a whole (Midgett and Doumas 2016). The term bullying describes repeated acts of physical, verbal, relational, or electronic abusive behavior (i.e., cyberbullying) that are perpetrated by individuals or groups against less powerful peers in an effort to cause physical or psychological harm (Nansel et al. 2001). The fact that bullying tends to increase in severity during late elementary school highlights the need for counselors in these schools to offer antibullying programs specifically tailored to this age group (Midgett and Doumas 2016). When it comes to controlling bullying in educational institutions, the school climate is a well-researched topic. In addition to sensitization, bystanders, according to most of the research, are the largest group involved in bullying behavior at school (Lambe et al. 2019). Thus, bystanders or peers should be the primary target of any intervention aiming to change the school climate. It is imperative for the school to participate in prevention programs in order to engage not only adult authorities but also other students.

While school-wide interventions that address bullying comprehensively have been proven effective, many schools, especially in low-to-middle income countries like Pakistan, lack the necessary resources to implement them (Sivaraman et al. 2019). Therefore, comprehensive antibullying bystander programs that train students to become peer advocates are a promising and cost-effective alternative for intervening in instances of bullying (Menesini et al. 2018; Midgett and Doumas 2016; Zambuto et al. 2020; Zhao and Chang 2019). Peer training can be conducted among students of the same age group or among students of different ages. Through this process, students learn from each other in an organized way. Peer training creates an opportunity for students to utilize their knowledge and experience in a meaningful way. This method of tutoring enables both trainer and trainee to gain self-confidence; the tutor by witnessing his or her self-competence in their ability to help someone, and the trainee by receiving positive reinforcement from their peers (Ali et al. 2015). It is recommended that along with teachers' professional development, components for peer training should also be introduced to reduce bullying incidents in educational setups.

The first intervention of its kind, the Sohanjana Antibullying Intervention, was created to address bullying in Pakistan's educational institutions through teachers' professional development and peer training for students to implement antibullying policies in a contextualized manner. The goal of peer training is to empower students to take an active role in creating a safe and respectful learning environment, and to promote a culture of kindness and inclusion. This study aims to examine the effects of a pilot peer-training module by implementing and evaluating it in four schools in Pakistan. One objective of the study focuses on raising the awareness of the peer mentors, who attended a four-hour training session. A second objective is to examine the impact of these mentors' use of their acquired knowledge to improve the school environment. In addition to the results from the pilot evaluation, we also present an overview of the module's content and activities. Additionally, the study discusses limitations and provides recommendations for future implementation.

## 2. Theoretical Background

### 2.1. The Bystander Intervention Model in Bullying

In bullying, bystanders are individuals who are neither victims nor perpetrators, but witnesses to bullying in person or online. A bystander can be a friend, a student, a peer, a teacher, a school staff member, a parent, a coach, or some other adult serving youth. Even strangers can be bystanders in cyberbullying situations (Bystanders to Bullying 2018). The behavior of student bystanders who witness a bullying incident can vary, and these behaviors have been classified into distinct bystander roles. These roles include "assistants", who actively assist the bully in victimizing the target; "reinforcers", who laugh or simply observe the situation; "outsiders", who remain neutral and often choose to disengage or walk away from the group; and "defenders", who step in to intervene and/or offer support to the target of bullying (Salmivalli et al. 1996, p. 15).

For more than 40 years, social psychology research has focused on bystander behavior. This was initially sparked by the outcry over the violent murder of Kitty Genovese in 1964 that was witnessed by bystanders who did not offer any help. It was proposed that the bystanders were reluctant to act due to either uncertainty about the emergency situation (known as pluralistic ignorance) or the belief that other people would take responsibility for helping the victim (known as diffusion of responsibility) (Nickerson et al. 2014). Thus, the presence of others can hinder helping behavior, which is known as the bystander effect (Dovidio et al. 2017). One explanation for the diffusion of responsibility is that an individual may feel less obligated to act when in a group rather than alone, potentially due to audience inhibition or the fear of being embarrassed in front of others (Latané and Nida 1981).

Latané and Darley's (1970) model for bystander intervention outlines a series of five sequential steps that must be taken in order to act: (a) noticing the event, (b) interpreting the event as an emergency in need of assistance, (c) assuming responsibility for intervening,

(d) knowing how to intervene or offer help, and (e) implementing the decision to intervene. While it has been reported that bullying is widespread in educational institutions in Pakistan, students may not always be aware of the relevant cues (e.g., hearing derogatory names) or may not perceive them as significant events (Saleem et al. 2021; Siddiqui et al. 2023; Nickerson et al. 2014). Once an event has been noticed, the situation must be correctly interpreted as an emergency requiring help. Research has shown that errors in decision-making can occur at this stage as individuals look to other bystanders to guide their interpretation. Additionally, ambiguity in the situation may make it difficult for an individual to recognize that help is needed, leading them to rely on cues from others to determine whether or not the situation is an emergency (Nickerson et al. 2014). After noticing an event and correctly interpreting it as an emergency requiring assistance, the next crucial step is for the individual to assume personal responsibility for intervening. In the presence of bystanders, people may believe that someone else will act or that responsibility (or blame) will be diffused throughout the group (Latané and Darley 1968). The following step involves knowing the appropriate actions to take in order to respond effectively to the situation in need of help. However, a lack of intervention skills (Burn 2009) can impede the ability to know which actions are necessary to intervene effectively in the situation. Ultimately, the final step in the bystander intervention model is to intervene in the situation.

Bystanders are often the initial witnesses to incidents and typically report them to teachers, which is why they are frequently trained in most interventions (Bjereld 2018). However, in Pakistan, the number of bystanders who inform a teacher of an incident or indirectly intervene appears to be low (Siddiqui et al. 2023). Gordon (2019) identified several reasons why bystanders may not intervene or report incidents to adults, including fear of victimization for reporting or intervening, a lack of knowledge about what to do in such situations, mistrust of adults, being taught to avoid such situations, and moral disengagement beliefs. It is essential to investigate the primary reasons for bystanders' silence in Pakistan. Furthermore, future intervention designs for educational institutions in Pakistan should emphasize the role of bystanders in the bullying chain. When students see their peers taking a stand against bullying, they are more likely to follow suit. This creates a positive environment where students are encouraged to act in a responsible and respectful manner towards one another. Moreover, when students are given the opportunity to take an active role in preventing bullying, they are more likely to feel invested and engaged in the process. This can lead to more buy-in from the student body and an increased likelihood of success.

## 2.2. Peer-Supported Interventions in Pakistan

Studies by Ahmed et al. (2022); Perveen et al. (2022); Shahid et al. (2022); and Siddiqui et al. (2021) have shown that bullying and cyberbullying are pervasive in Pakistani educational institutions and can have detrimental effects on the physical and mental well-being of students. Previous research has also indicated that importing or implementing antibullying interventions from other countries to combat bullying in Pakistani educational institutions has not always been effective (McFarlane et al. 2017; Siddiqui et al. 2023). Victims usually of adolescent age are reluctant to share their problems with adults. They value their privacy and seek anonymous help through peer support (Jacobs et al. 2016; Matuschka et al. 2022). It is also reported that youth do not involve adults concerning victimization issues because of distrust of adults and concerns about being blamed (Bjereld 2018). Most of the time, children do not disclose bullying incidents as they feel ashamed about being a victim (ibid). Even with encouragement, many bullied students do not disclose or involve adults in their difficulties (Black et al. 2010) and often contact peers for help. Peer-led programs are being developed in different regions where peers receive training sessions so that they can participate in school-wide events aimed at reducing bullying (Espelage and Hong 2017). A qualitative study conducted by Tzani-Pepelasi et al. (2019) in the UK explored the effectiveness of the 'buddy approach' in improving the overall school environment. The study shed light on the positive impact of this approach on

both young mentors and mentees. Semi-structured interviews with open-ended questions were used to explore the effectiveness of the buddy program approach. The results showed that the program was successful in supporting students to develop friendships, as well as feelings of safety, belonging, and protection. It also helped to develop a sense of responsibility, satisfaction, and pride. Similarly, a qualitative and comparative study by Biswas et al. (2020) found that peer-support programs in both Asian and African regions were associated with a reduced risk of bullying victimization. Another quasi-experimental study conducted by Ferrer-Cascales et al. (2019) with 2057 Spanish students evaluated the effectiveness of the TEI program, an intervention based on peer tutoring aiming to reduce bullying and cyberbullying and improve the school climate. The results of the study showed a significant decrease in bullying behavior, peer victimization, fighting, cyberbullying, and cybervictimization within the experimental group following the implementation of the intervention. In addition, the experimental group showed significant improvements in various aspects of the school climate. This study serves as an illustrative example of the considerable potential of peer antibullying training to effectively reduce bullying and victimization.

Based on the success rate of peer involvement in antibullying interventions (Biswas et al. 2020; Ferrer-Cascales et al. 2019; Menesini et al. 2018; Tzani-Pepelasi et al. 2019; Zambuto et al. 2020), the Sohanjana Antibullying Intervention is also designed to involve high-status peers to help school administrators and teachers to create a zero-bullying environment. Students are more likely to listen to and trust their peers than they are to listen to authority figures such as teachers or administrators; having peers lead intervention efforts can lead to a more effective dissemination of information and a greater impact on changing student behavior. Unfortunately, no such training programs are reported in Pakistan's context. There is a great deal of concern about bullying among children and adolescents in Pakistan, but it is rarely addressed and little research has been conducted (Ashfaq et al. 2018). While certain researchers have reported interventions where external practitioners implemented antibullying strategies to address bullying in educational institutions (Karmaliani et al. 2020; Maryam and Ijaz 2019; McFarlane et al. 2017), there have been no reports in Pakistan's context of training peers to combat this issue and cultivate prosocial bystanders. Similarly, there is need to expand research on bullying and interventions in low-to-middle income countries like Pakistan (Sivaraman et al. 2019).

## 3. Summary of the Components of the Peer-Training Module

To emphasize the importance of bystanders in the bullying cycle, the Sohanjana Antibullying Intervention introduced a peer-training module into Pakistan's educational institutions for the first time. The Sohanjana Antibullying Intervention Peer-Training Module serves the crucial function of familiarizing students with general awareness-raising techniques. This involves educating them about the various roles of participants and group dynamics involved in bullying. According to Salmivalli's (1999) findings, clarifying different participant roles to students reinforces the gravity of the situation and helps them to realize that their behavior, even if unintended, could have encouraged bullying. Information on group mechanisms provides insight into why individuals may behave in ways that contradict their true intentions, sometimes without even realizing it. In addition, students were provided with online and hands-on training to help them create safe social media accounts. Hands-on activities involve all students manipulating objects by hand or practicing whole things with the touch of their hands and has been proven as a successful technique for teaching students new learning (Nurjanah et al. 2021). Students also participated in role play activities where they practiced active, passive, and prosocial bystander roles. Incorporating realistic, or real-world, problems into role play enhances content skills, as well as skills necessary for future success (Clapper 2010). Real-life bullying cases reported by students were also used to help students analyze situations and suggest how they could react in similar situations. According to Yin (1994), case studies allow learners to gain a deeper understanding of individuals, organizations, and social and political phenomena.

A case study enables an investigation to retain the holistic and meaningful characteristics of real-life events (p. 14), including organizational processes. A part of the training was aimed at helping schools introduce antibullying policies. Peer training consists of training students to lead antibullying campaigns within their school communities. This can include creating posters, organizing events, and spreading awareness about the impact of bullying and the importance of kindness and respect. During the training module, participants were engaged in discussions to help them understand how victims of bullying can be encouraged to stand up to bullying and perpetrators can be encouraged to engage in more prosocial activities. During this part of the training, students were trained to support and encourage each other. The goal was to create a positive, supportive community where students can turn for help and feel valued and accepted. Different aspects of the Sohanjana Antibullying Intervention Peer-Training Module are shown in Figure 1 and Table 1.

**Figure 1.** Summary of the Sohanjana Peer-Training Module.

**Table 1.** Summary of content and activities.

| Content of the Workshop | Supporting Literature | Activities |
|---|---|---|
| Definition of bullying and its subtypes | Burger et al. (2015); Grigg (2010); National Center Against Bullying (NCAB) (2021); Siddiqui et al. (2021) | Brain storming to help students understand examples of cyberbullying |
| Importance of identification of bullying and its consequences | Asif (2016); Burkhart and Keder (2020); Inamullah et al. (2016) | Video presentation on the suffering of a victimized child, followed by discussion |
| Importance of bystander intervention and types of bystanders | Arënliu et al. (2020); Lambe et al. (2019); Polanin et al. (2012) | Role play using Dan Olweus' picture of the bullying cycle |
| Why victims remain silent and why peer mentors should be trained | Bjereld (2018) | Discussion |
| Identification of victimization and perpetration behaviors | Burkhart and Keder (2020); Davis and Nixon (2010); Guo (2016); Hastings and Bham (2003); Hutzell and Payne (2018); Nas et al. (2008); National Autistic Society (2017); Sampasa-Kanyinga et al. (2014); Van Geel et al. (2014); Warning Signs of Bullying and What to Do (2018) | Group Activity—Case studies of bullying incidents to make suggestions for how students can act in a similar situation |
| Developing problem solving skills | Cook et al. (2010); Hall (2006) | |
| What to do when bullying victimization is noticed | How to Identify Cyberbullying (2018); Warning Signs for Bullying (2021) | Motivational video on the importance of standing up to bullying followed by discussion |
| Introduction to legal practices and law enforcement agencies in Pakistan to control cybercrime | Is It Harmless Trolling or Cyberbullying? (2020) | Hands-on activity to change Facebook profile settings *. Handouts and short presentations on technological possibilities and safety features |
| How to support the school in implementing an antibullying policy How to support victims and confront perpetrators | Analitis et al. (2009); Beane (2008); British Association for Counselling (1991); Bullying Bystanders . . . Become Upstanders (2022); Bystanders to Bullying (2018); Carney and Merrell (2001); Cook et al. (2010); Dake (2021); Eisenberg et al. (1997); I'm a Bystander to Bullying. How Can I Offer Support? (2022); Kochenderfer and Ladd (1997); Kupchik and Farina (2016); Maines and Robinson (1992); Midgett (2016); Olweus (2004); Pikas (2002); Rigby (2010); Rigby and Griffiths (2011); Rigby (2011); Sherer and Nickerson (2010); Simons et al. (2004); Vernetti (2022) | Each mentor prepares a speech and presents their motto and the upcoming activities they will do to make the school environment safe Introduction to no-blame approach, shared concern method, stealing the show, turning it around, accompanying others, coaching compassion |

* Additional information sheets were provided for TiKToK and Instagram, but due to time constraints, only Facebook settings were discussed, as FB is still the most common social network, as reported by Siddiqui and Schultze-Krumbholz (2023).

## 4. Research Methodology

In this study, a mixed-method approach was used. The quantitative aspect of the study utilized a pre-post survey design to evaluate the impact of the training on enhancing participants' comprehension of bullying concepts and improving their responsibility for antibullying strategies in their respective facilities. This section also included closed-ended questions to gather feedback on the different training components. Furthermore, we utilized open-ended questions in the qualitative part to investigate the strengths, limitations, and suggestions for future training programs. The training was conducted in the form of online workshops via the Zoom platform and involved participants from four educational institutions (two from urban school settings—Schools A and B—and two from rural schools—Schools C and D) who took part in both the workshops and pre- and post-tests.

To assess the impact of the peer-mentor training on improving the school environment and reducing incidents of bullying and victimization, the researchers used a pre-post questionnaire. It is important to note that although four schools initially participated in the workshop, only three of them (School A, B, and C) continued to implement the antibullying program and provided pre- and post-workshop data.

### 4.1. Sample

To test the peer-training module, we reached out to numerous educational institutions in Pakistan and invited them to participate in a four-hour online workshop. Invitations were only extended to institutions that had previously taken part in the 32 h Sohanjana Antibullying Teachers' Professional Development Program. These invitations were sent to teachers who had attended the eight-day workshop and were already familiar with the objectives of the Sohanjana Antibullying Intervention. The invitation email included information about the authors' affiliation, the study's aims, and the workshop topic. As part of the initiative, peers were encouraged to collaborate with educators to combat bullying in their respective institutions. Teachers who expressed interest in the training received additional information, such as the training date, participant capacity, content, and activity details. Teachers informed their students and parents about the training, and interested students signed up and participated in the session. Participants who attended the workshop and completed the pre-post questionnaires received a certificate of participation. The peer-training workshop involved 38 students (boys = 26, girls = 12) aged 10 to 15 (mean age = 11.92, SD = 1.477) from four schools (two rural, two urban). The demographics of the participants are shared in Table 2. To assess the impact of the peer-mentor training on improving the school environment and reducing incidents of bullying and victimization, prior to the workshop and program implementation, students from the participating schools provided questionnaire data. Then, three months after the workshop and program implementation, the students were asked to provide post-workshop responses. The demographic details of the students from these three schools are shown in Table 3.

**Table 2.** School * Grade cross-tabulation (peer mentors).

| | | Grades of Participants | | | | | | Total |
| | | 3rd | 5th | 6th | 7th | 8th | 9th | |
|---|---|---|---|---|---|---|---|---|
| Schools | School A (Urban) | - | - | - | - | 5 | 4 | 9 |
| | School B (Urban) | - | 7 | - | 5 | - | - | 12 |
| | School C (Rural) | - | 7 | 1 | - | - | - | 8 |
| | School D (Rural) * | 1 | 5 | 3 | - | - | - | 9 |
| Total | | 1 | 19 | 4 | 5 | 5 | 4 | 38 |

* Although School D participated in the online peer mentor workshop, they did not continue with the implementation of the program or provide pre-post data from their students.

**Table 3.** Demographics of students (N = 219).

| Variable | | N | % |
|---|---|---|---|
| Gender | Male | 152 | 69.4 |
| | Female | 67 | 30.6 |

**Table 3.** *Cont.*

| Variable | | N | % |
|---|---|---|---|
| Grades | 3rd | 3 | 1.4 |
| | 4th | 5 | 2.3 |
| | 5th | 74 | 33.8 |
| | 6th | 3 | 1.4 |
| | 7th | 51 | 23.3 |
| | 8th | 47 | 21.5 |
| | 9th | 36 | 16.4 |
| Participants from each school | School A (urban setting) | 83 | 37.9 |
| | School B (urban setting) | 107 | 48.9 |
| | School C (rural setting) | 29 | 13.2 |
| Total | | 219 | 100% |

*4.2. Instruments*

The authors developed a range of open and closed-ended questions for the peer-training module based on the workshop content (see Figure 1 for an example question). The content of the workshop was used as a basis for the development of a set of questions and corresponding rubrics, including sample responses or model responses. These questions were administered to participants both before and after the program, and their responses were scored using these rubrics. A team of researchers reviewed these questions in detail before conducting the workshop. The open-ended questions were designed to explore different aspects of the workshop and to gather suggestions for improvement. The researchers carefully considered these suggestions and incorporated them into future workshops for ongoing improvement. Google forms were utilized to create online questionnaires for data collection in both the pre-test and post-test. The pretest link was shared at the beginning of training. At the conclusion of the workshop, the link for the post-test was shared. To assess the contents, activities, and delivery of the workshops, both open-ended and closed-ended questions were utilized. Closed-ended questions, such as "Did you enjoy attending this session?" were answered using a 3-point Likert scale (Yes, Partially, No), while open-ended questions, such as "How could the workshop have been improved?" were also included. These open-ended questions were designed to explore ways to enhance various aspects of the workshops. The open-ended feedback responses consisted of single words (e.g., "too long") or brief statements (e.g., "duration should be reduced"), which facilitated the researchers' analysis of common themes and subsequent reporting of descriptive statistics. The researchers carefully scrutinized the suggestions provided and discussed them in detail during their discussions and recommendations. The researchers then examined both the outcomes and suggestions provided by participants through a combination of open-ended and closed-ended feedback questionnaires.

Specific items from the Revised Olweus Bully/Victim Questionnaire (Olweus 1996) were used to determine the impact of peer-mentor training on the overall improvement of the school environment. This questionnaire is a recognized tool for assessing incidents of bullying, covering both the role of the perpetrator and the role of the victim. Although the questionnaire consists of a total of 40 questions, the researchers focused specifically on items related to improving the role of bystanders and fostering students' confidence in seeking help from adults to address the problem. The Likert scale used in this questionnaire varies for different statements and consequently each statement within the analysis provides an explanation of the specific Likert scale used. A Google form link was created for each school, with separate forms for pre and post scores, and shared with the schools' management. Prior to the program's implementation, consent was obtained from parents, who agreed and returned the consent forms with their signatures, granting permission for the school to employ the training and collect data from the students. Moreover, in both the pre and post questionnaires, students were once again asked to confirm that their parents were informed and had given consent for their participation in the survey. School B and School

C provided computers and devices for their students, who filled out the questionnaires during school hours. On the other hand, in School A, the link was shared with students and their parents through the school portal and email, and they completed the forms at home in the presence of their parents. Participation in the survey was voluntary, and only students who were willing to share their data completed the questionnaire.

## 5. Data Analysis and Results

The online pilot workshops were evaluated based on the participants' understanding of bullying concepts, the consequences of bullying, and strategies to control bullying intervention through a Wilcoxon sum-of-means test, split plot ANOVA, and descriptive statistics using IBM SPSS Statistics version 27 Armonk, New York, USA (the license of the Software was provided by Technische Universitat Berlin to authors). Descriptive statistics were used to evaluate the feedback from participants, and Wilcoxon sum-of-means values were calculated due to the small sample sizes for pre-post differences to evaluate participants' knowledge about the content of the workshop related to bullying and peer roles in the bullying cycle. In order to evaluate the overall training impact on improving the school environment, paired sample *t*-tests were used to compare pre- and post-workshop data, providing insight into the statistical significance of the findings. Descriptive statistics were also used to summarize and present the findings in a comprehensive manner. In addition, thematic analysis was used to analyze the responses to the open-ended questions, enabling the identification of common themes and patterns within the qualitative data. Thematic analysis is a qualitative research method used to identify and analyze patterns, themes, and meanings within a dataset. It involves the systematic organization and interpretation of textual or qualitative data to reveal underlying themes or patterns of meaning.

### 5.1. Descriptive Statistics (Peer Training)
Closed and Open-Ended Feedback Results

In addition to inquiring about participants' comprehension of bullying and intervention, both open-ended and closed-ended questions were posed to assess the strengths and drawbacks of the workshop through descriptive statistics. Descriptive statistics are typically used to summarize and describe the main features or characteristics of a dataset. They are often used in data analysis to provide a clear and concise understanding of the outcomes, often as a preliminary step before more advanced statistical techniques are applied. Descriptive statistics help to organize, summarize, and present data in a meaningful way, allowing patterns, trends, and key insights to be identified. The results of the closed-ended feedback responses showed that 74% of participants were highly satisfied with the workshop's content, activities, and instructions, and would rate it as excellent (refer to Table 4). Furthermore, 91% of participants expressed enjoyment of being part of the session (refer to Table 4). After the workshop, open-ended questions were also posed, and certain participants mentioned the extended duration of the workshop (4 h with a 20 min break each time) as a limitation. In the open-ended feedback section, a significant number of participants chose not to provide any suggestions or limitations, simply stating "everything was good" or "no comment". However, a subset of respondents provided concise statements that included suggestions and limitations. These statements were carefully analyzed by the researchers, who used thematic analysis to categorize and report on the responses.

**Table 4.** Feedback (descriptive statistics).

| Feedback Questions and Responses | | | | |
|---|---|---|---|---|
| This workshop was | Excellent 74.4% | | Good 25.6% | |
| Did you enjoy attending this session? | Yes 90.7% | Partially 9.3% | No ----- | |
| What did you like about the workshop? Multiple responses were accepted and reported. | Discussion about Bullying 20 participants | Way of Instruction 12 participants | Activities 12 participants | Content of the Workshop 5 participants |
| What did you not like about the workshop? Multiple responses were accepted and reported. | Duration was long 10 participants | Online Workshop/Network Issues 3 participants | Sometimes discussion was beyond topic 2 participants | Not everybody got a chance to share their speech 1 participant |
| Suggestions Multiple responses were accepted and reported. | More explanation of activities 1 participant | Longer break—20 mins was not enough 1 participant | | Duration should be decreased 3 participants |

*5.2. Pre-Post Comparisons (Peer Training)*

5.2.1. Change in Knowledge and Understanding

The research analysis aimed to determine the difference in participants' knowledge about the workshop contents before and after attending the workshop. The Wilcoxon sum-of-means test was used to assess these differences with a fixed $p$-value of 0.05. The Wilcoxon test is a non-parametric statistical test that is utilized instead of the paired sample $t$-test when the data are not normally distributed or the sample size is very small (Dwivedi et al. 2017; Wilcoxon 1992). As the sample size for the peer-mentor training was 38, the Wilcoxon test was preferred. The analysis revealed a significant improvement in participants' knowledge and understanding as the post-test participants provided more correct responses as compared to the pre-test with a positive difference observed among 30 out of 38 participants (see Table 5). The obtained $p$-value of 0.00 (<0.05) and the significant increase in mean scores from 3.6316 to 5.1579 provide clear evidence of improved scores in the post-workshop test. These results indicate that participants demonstrated a greater number of correct answers specifically related to the workshop content. This reflects an improved understanding of concepts such as bullying, the role of bystanders, the responsibilities of peer mentors in implementing antibullying policies, and knowledge of how to intervene and guide other students during bullying incidents.

**Table 5.** Wilcoxon sum-of-means test.

| Module | Pre-Post Test Sum | Mean Value | Percentiles Median (50th) | Standard Deviation | Wilcoxon (Z-Value) | *p*-Value | Frequencies (Post-Test Sum-Pre-Test Sum) | N |
|---|---|---|---|---|---|---|---|---|
| Peer-Training Module | Pre-test Sum | 3.6316 | 3.0000 | 1.96484 | −4.720 [a] | 0.000 | Negative Differences [a] | 2 [b] |
| | Post-test Sum | 5.1579 | 5.0000 | 1.46170 | | | Positive Differences [b] | 30 [c] |
| | | | | | | | Ties [c] | 6 [d] |

[a]. Based on negative ranks; [b]. Post-Test Sum < Pre-Test Sum; [c]. Post-Test Sum > Pre-Test Sum; [d]. Post-Test Sum = Pre-Test Sum.

### 5.2.2. Comparison of School Settings (Peer Training)

In order to evaluate differences among pre- and post-test results from rural and urban populations, a split plot ANOVA test was conducted. The split plot ANOVA method is used to determine whether two or more repeated measures from two or more groups significantly differ from each other on a variable of interest. It is recommended that the variable of interest should be continuous, be normally distributed, and have a similar spread across the groups. It is evident from the graph diagram (see Figure 2) that the training has improved knowledge and understanding about bullying among participants from both the rural and urban groups. However, the change is more evident in the rural group, as their initial knowledge about the topic was less than that of the groups from urban settings. The mean scores showed that urban students (mean = 4.908, SD = 1.513) had more initial knowledge than rural students (mean = 2.059, SD = 1.144). Likewise, the post-training scores of the urban group (mean = 6.095, SD = 0.995) were better than those of rural participants (mean = 4.000, SD = 1.060). However, although there were differences in the improvements observed between rural and urban areas, these differences did not reach statistically significant levels (refer to Table 6).

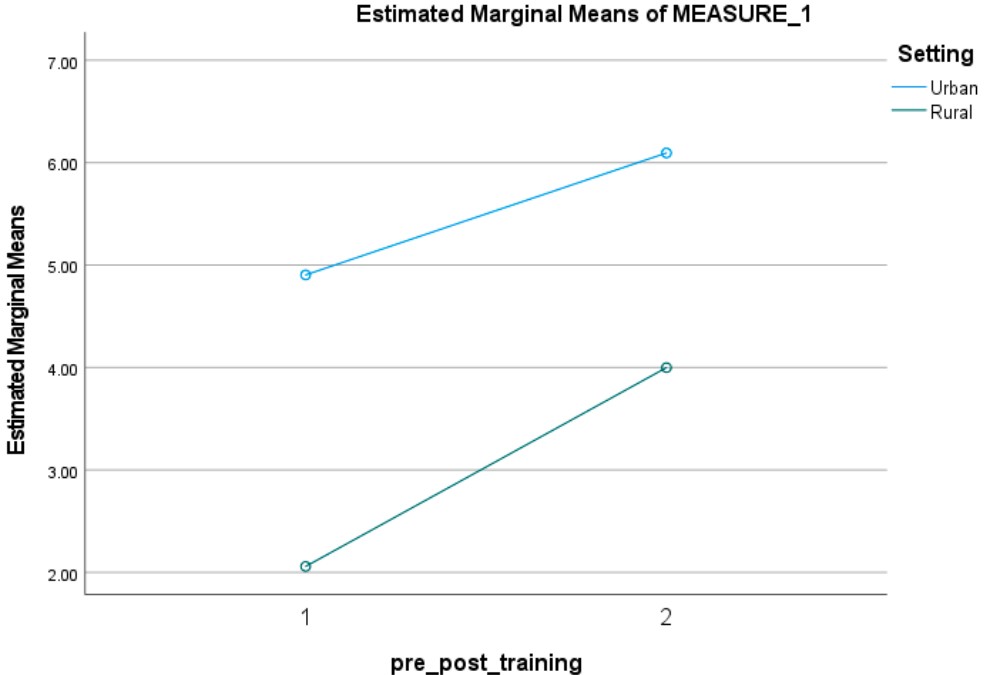

**Figure 2.** Pre-post training differences among participants from urban and rural school settings.

**Table 6.** Pre-post comparisons (split plot ANOVA).

| Variable | Pre-Post | Gender | Mean | Std. Deviation | N | Df | Mean Square | F | Sig. |
|---|---|---|---|---|---|---|---|---|---|
| Impact of Peer Training on Participants Knowledge | Pre-Test Score | Urban | 4.908 | 1.513 | 21 | | | | |
| | | Rural | 2.059 | 1.144 | 17 | 1 | 2.647 | 3.276 | 0.079 |
| | Post-Test Scores | Urban | 6.095 | 0.995 | 21 | | | | |
| | | Ural | 4.000 | 1.060 | 17 | | | | |
| Impact of Peer Training on Participants Knowledge | Pre-Test Score | Boys | 3.615 | 2.063 | 26 | | | | |
| | | Girls | 4.000 | 1.758 | 12 | 1 | 0.327 | 0.374 | 0.545 |
| | Post-Test Scores | Boys | 5.077 | 1.547 | 25 | | | | |
| | | Girls | 5.333 | 1.302 | 12 | | | | |

### 5.2.3. Gender Differences

Similarly, to identify gender differences between pre- and post-training results, a split plot ANOVA test was conducted. The results showed that despite a smaller number of girls participating in the training, the initial (N = 12, mean = 4.000, SD = 1.758) and final scores (mean = 5.33, SD = 1.3026) of girls were slightly better than boys' initial (N = 26, mean = 3.440, SD = 2.103) and final scores (mean = 5.077, SD = 1.547) (see Figure 3). The results of this study indicate that girls perceive and understand the topic better than boys. However, due to differences in sample sizes (boys = 26 and girls = 12), these findings cannot be generalized to the entire population. Moreover, although there were differences in the improvements observed between male and female participants, these differences did not reach statistically significant levels (refer to Table 6).

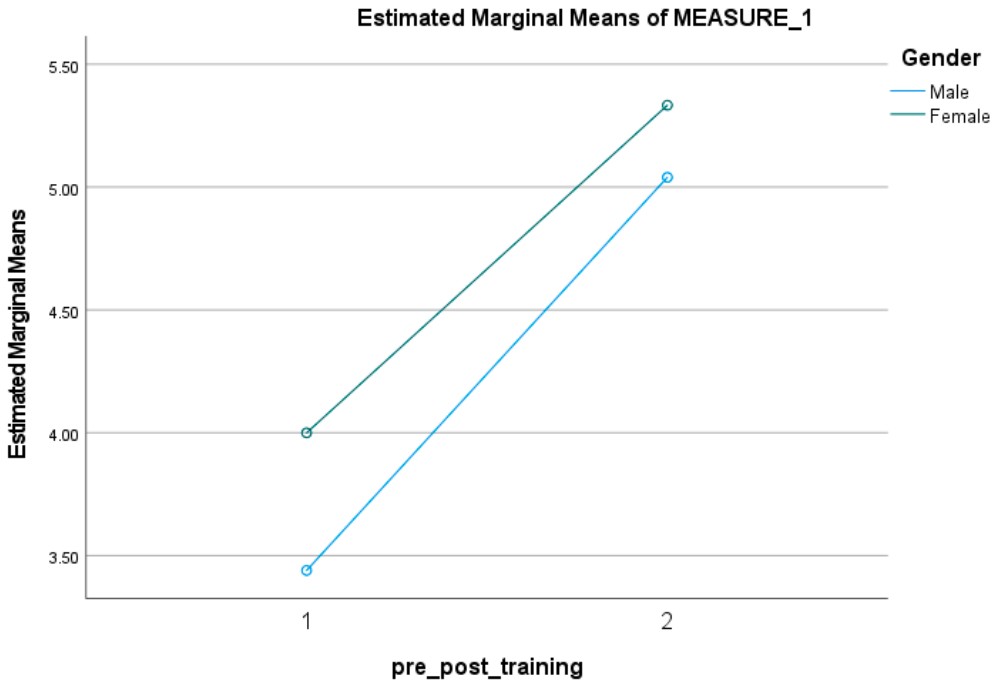

**Figure 3.** Pre-post training differences among male and female participants.

### 5.3. Evaluation of Workshop on School Environment

Following their participation in the workshop, the peer-mentor team actively participated in various school-wide events and applied their newly acquired knowledge by working with trained teachers to implement antibullying strategies in the school. They took the initiative to introduce themselves to their peers through classroom visits and morning assemblies and actively invited their peers to participate in the school's antibullying program. In addition, the peer mentors were trained to support both victims and

perpetrators of bullying. To meet the needs of students who were reluctant to openly share their experiences or seek help from adults, the peer mentors shared their contact details to offer support. This implementation phase lasted for three months. The collection of data from students (N = 219) was conducted at two timepoints: before peer-mentor training and again three months after the training was conducted.

In order to assess the impact of the intervention on different aspects of the school climate, specific items from the Olweus Bully/Victim Questionnaire were adopted. These items assessed the frequency of bullying incidents, the role of peer bystanders in intervening, the dynamics of groups involving the bullied students, the extent to which victimized students sought help from others, the willingness to offer help to fellow students, and empathy towards victimized students. A paired sample *t*-test was used to examine the differences between the pre- and post-test scores (refer to Table 7). Descriptive statistics were also used to assess the effectiveness of the program. Students were also given an open-ended question to comment on the program, which allowed for multiple responses. These responses were subjected to thematic analysis, resulting in the identification of three distinct themes: appreciation of the program, criticism and suggestion for the intervention, and changes observed in peers and self-reflection.

Analysis of the pre- and post-intervention data collected from students using items from the Olweus Bully/Victim Questionnaire revealed a decrease in post-victimization and perpetration scores, as well as an improvement in students' antibullying attitudes. The mean scores related to the statement regarding students' attitudes towards sharing their suffering showed a decrease in the post-intervention data, which serves as an indication of reduced victimization as a result of the implementation of the program. On further examination, it was found that in the pre-intervention data, four students reported that they did not confide in anyone about their experiences of bullying. This number decreased to two students in the post-intervention data. Encouragingly, there was an increase in the number of students who began to trust and share their bullying experiences with teachers, rising from 18 to 21. Similarly, reporting to parents/guardians increased from 19 to 25 and to siblings from 2 to 7. However, sharing with friends decreased from 26 to 21, which is an area for further research and investigation. A reduction in group bullying was observed, indicating a decrease in the number of pupils involved in bullying groups. Prior to the intervention, when students were asked about the number of individuals who bullied them, 27 students reported being bullied by a group of 2–3 students, while 11 students reported being bullied by a group of 4–9 students. After the intervention, these numbers decreased to 18 and 10 respectively, indicating a shift in group dynamics. This suggests that the number of bystanders who had previously joined the bullies to keep them company had decreased. It is important to note, however, that the results cannot be considered robust as the necessary level of significance was not achieved. Nevertheless, the significant improvement in peer/bystander attitudes is a notable finding. Overall, these data suggest that the training of peer mentors and the implementation of the intervention significantly improved bystander intervention in bullying incidents (refer to Table 7).

**Table 7.** Pre-post intervention results (paired sample *t*-test).

| Variable | Statement | Pre-Post | N | Mean | Std. Deviation | t-Value | df | Sig. 2-Tailed |
|---|---|---|---|---|---|---|---|---|
| Bullying Victimization 5-point Likert Scale (1 = I haven't been bullied at school in the past couple of months, 5 = several times a week) | How often have you been bullied at school in the past couple of months? | Pre-intervention | 219 | 1.662 | 1.020 | 1.302 | 218 | 0.194 |
| | | Post-intervention | 219 | 1.557 | 1.009 | | | |
| Bullying Group Dynamics 5-point Likert Scale (1 = I haven't been bullied at school in the past couple of months, 2 = mainly by 1 student, 5 = by a group of more than 9 students) | By how many students have you usually been bullied? | Pre-intervention | 219 | 1.648 | 1.112 | 1.296 | 218 | 0.196 |
| | | Post-intervention | 219 | 1.5342 | 1.010 | | | |
| Changing attitude towards sharing about victimization 3-point Likert Scale (1 = I haven't been bullied at school in the past couple of months, 2 = I have been bullied, but I have not told anyone, 3 = I have been bullied and I have told somebody about it) | Have you told anyone that you have been bullied in the past couple of months? | Pre-intervention | 219 | 1.5114 | 0.8147 | 0.314 | 218 | 0.729 |
| | | Post-intervention | 219 | 1.4886 | 0.79771 | | | |
| Peer/Bystander Intervention 5-point Likert Scale (1 = almost never, 5 = almost always) | How often do other students try to put a stop to it when a student is being bullied at school? | Pre-intervention | 219 | 2.6667 | 1.29690 | −2.537 | 218 | 0.012 |
| | | Post-intervention | 219 | 2.9589 | 1.40874 | | | |
| Bullying Perpetration 5-point Likert Scale (1 = I haven't bullied another student(s) at school in the past couple of months, 5 = several times a week) | How often have you taken part in bullying another student(s) at school in the past couple of months? | Pre-intervention | 219 | 1.1826 | 0.6084 | 0.646 | 218 | 0.519 |
| | | Post-intervention | 219 | 1.10507 | 0.4694 | | | |
| Changing attitudes towards bullying 6-point Likert Scale (1 = Yes, 6 = Definitely No) | Do you think you could join in bullying a student whom you didn't like? | Pre-intervention | 219 | 4.8950 | 1.51534 | −1.074 | 218 | 0.284 |
| | | Post-intervention | 219 | 5.0137 | 1.21715 | | | |

*5.4. Analysis of Open-Ended Questions (Post-Intervention Implementation Results)*

During the open-ended section, where students were encouraged to share any aspects not covered, a variety of responses were collected. Certain students expressed how they had seen changes in themselves and their peers, while others offered suggestions and even criticism. Many students also praised the efforts and appreciated the program.

5.4.1. Self-Reflection and Observation

One student shared a personal experience that illustrates the positive impact of the program. He mentioned how, after the implementation of the program, he stood up for his shy friend in Year 4 who was being bullied. Describing the situation, the student stated:

*"I've written about how teachers don't fully address problems because some students don't even speak up. In my friend's case, I confronted the bullies and informed an adult and the problem was solved."*

One other student shared her experienced:

*"My best friend has stopped making derogatory comments directly to my face, although she still occasionally makes comments about my weight and appearance. However, these instances are relatively rare."*

Several students shared their self-reflections and the changes they had observed. Below are some further examples:

*"I intervened and stopped my classmates from bullying by firmly telling them to stop and reporting the incidents to the teacher. This experience was very helpful because it made me realize how important it is to prevent others from being bullied. Even though it may be difficult or uncomfortable, I now understand that I should try to help them no matter how long the bullying has been going on."*

*"I have not personally experienced bullying, nor have I bullied anyone. However, if I witness someone being bullied in the future, I am determined to intervene and put a stop to it."*

*"We must stand up against bullying, no matter what it takes."*

*"I have been actively involved in my school's antibullying campaign, taking part in various activities to raise awareness and promote a bully-free environment."*

5.4.2. Praising Efforts and Appreciation

Several students expressed their appreciation for the team and the peer mentors and made positive statements such as:

*"The effort put into this initiative was commendable."*

*"The antibullying campaign has had a noticeable impact in our school."*

*"I am very much against bullying and I think the teachers and bully-busters have done us a great service. Keep up the good work."*

*"With all the adults now aware of bullying, it feels like bullying is finally coming to an end."*

5.4.3. Critics and Suggestion

Certain students expressed criticism and made suggestions to improve the program. Below are two example statements:

*"I spoke to my teacher with ten other girls about bullying in the gym. Although she talked to the boys involved, nothing seems to have changed. So, I don't think this program will make a difference".*

*"Sometimes our teachers don't do much when students are being bullied. Although the issue is discussed, the problem of cyberbullying is still prevalent in today's world".*

To address this issue, a comprehensive cyberbullying prevention training has also been developed for teachers' professional development.

> *"I wish that bullying would stop for all students and that there would be someone to support me when I am being bullied."*

> *"This program should be mandatory for teachers and support staff as they are often the ones who can intervene and stop bullying."*

> *"Teachers find it difficult to control students in Grades 10 and 11, so stricter measures should be taken against bullying."*

In response to this comment, a comprehensive training program was developed for teachers and school administrators on the zero-tolerance antibullying policy.

> *"I recommend organizing discussion panels, speeches, assertiveness training and counselling sessions in every class at least twice a year from the very beginning to address bullying, harassment, self-esteem and respect."*

In response to this comment, a comprehensive training program was developed for teachers and school administrators on the counselling and effective communication. It will be recommended for future programs that these sessions should continue throughout the year at intervals.

> *"Setting up a helpline is great, but I doubt that a hurt and scared young child would feel comfortable contacting you. They may be too afraid and feel they have to deal with these issues on their own. Instead of surveys, consider showing examples of people who have been bullied and how they overcame their experiences. Bring in guest speakers who can share their personal stories and use real witnesses who have been through this terrible ordeal."*

In future programs, motivational speakers or students suffered or witnessed bullying will also be requested to be the part of the program.

> *"Bullying or teasing often takes place between lessons, when teachers take too long to reach the classroom and no teacher is aware of the situation. Victims are reluctant to report for fear of further bullying."*

## 6. Discussion

Research has indicated that bullying and cyberbullying are widespread in educational institutions in Pakistan and have adverse effects on students' physical and mental well-being (Ahmed et al. 2022; Perveen et al. 2022; Shahid et al. 2022; Siddiqui et al. 2021). Previous literature has demonstrated that adapting or adopting antibullying interventions from other countries to prevent bullying in Pakistani educational institutions has not always been successful (McFarlane et al. 2017; Siddiqui et al. 2023). As a solution, the Sohanjana Antibullying Intervention was developed specifically for Pakistani teachers and is the first contextualized antibullying intervention (Siddiqui et al. 2023). The program involves a 4 h session of training for students/peers and teachers who have undergone 32 h of training at the same institute. Its primary objectives are to increase peers' knowledge and enable them to identify bullying and victimization problems in educational settings, take swift action to assist students, and establish a supportive school environment. This study aims to present the preliminary results of an online peer antibullying training program's effectiveness and acceptance. An evaluation was also carried out to assess the overall impact of the implementation of antibullying policies by peer mentors and trained teachers on students' prosocial behavior and the overall improvement of the school environment. The researchers employed a design that enabled them to evaluate the module's impact on participants' knowledge and understanding. The Wilcoxon sum-of-means test revealed a notable improvement in participants' understanding related to bullying interventions in a peer-training session. Based on these findings, the authors concluded that the peer-training module of the Sohanjana Antibullying Intervention is suitable for preparing a team of peer mentors who can collaborate with school-wide antibullying events and contribute to

improving the overall school climate. The piloting of the peer-training module in preparing a peer mentor team was successful and could be implemented in the future for peer training in a similar manner.

It was observed that girls' involvement in the peer-training module was less than that of boys, indicating that Pakistani culture still does not encourage girls to take a stand against bullying or participate in such interventions. According to Hinduja et al. (2023), girls' education and participation in assertive activities are relatively low in patriarchal Pakistani societies due to stereotypical family roles, cultural and religious inclinations, and the portrayal of women as passive, dependent, and submissive (Islam and Asadullah 2018). Although the male-to-female ratio is almost the same (1033 males per 1000 females) (Pakistan Population 2023), girls have a much smaller role in educational activities, which is consistent with the findings of the current study. Despite their lower participation in the training, girls showed better results in the pre-post assessments, but due to the insignificant differences observed, these conclusions should be treated very carefully. Further research with a larger sample size is recommended to make robust conclusions.

The results of the split plot ANOVA indicated differences between students from urban and rural areas in terms of their initial and final knowledge of bullying. Tayyaba (2012) argues that educational institutions in rural and urban regions differ, which affects the academic performance of students. Students in urban areas tend to have a greater exposure to the media and, consequently, possess a higher level of initial knowledge. However, during post-training, there was a reasonable improvement in knowledge observed among students from rural regions as well. Nevertheless, due to the lack of achieving the significance threshold, these results should be treated with great care. However, it can be inferred that if students in rural areas are afforded equal educational opportunities to their urban counterparts, they have the potential to attain similar outcomes. To strengthen this claim, further research is recommended.

This study also aims to assess the impact of the training program on various aspects of the school environment, such as bullying perpetration, victimization, attitudes of students, bystanders, and respondents towards bullying, and more. The implementation of the Sohanjana Antibullying Intervention Peer-Training Module produced mixed results. While there were reductions in the levels of victimization and perpetration in terms of absolute numbers, these results did not reach statistical significance. Similarly, improvements in pupils' attitudes towards antibullying and reductions in group bullying were not statistically significant. Previous studies have suggested that both whole-school antibullying programs (Hensums et al. 2022) and targeted interventions (Johander et al. 2021) are generally less effective in adolescence than in childhood. Nonetheless, even one less student that is being victimized or taking part in bullying is a welcome development. Previous research has consistently highlighted the challenges of designing interventions for adolescents, as they undergo significant physiological and psychological changes during the transition to adulthood (Yeager et al. 2015). This period is characterized by increased rebelliousness and internal changes. Crone and Dahl (2012) identified the onset of puberty and the associated increase in testosterone levels as factors contributing to an increased desire for social status. Engaging in social aggression at this stage of development may reflect increased social motivation rather than dysfunction, as observed at earlier ages. However, it is important to note that explicit rules against certain social behaviors enforced by adults or peer leaders, along with direct instruction in the classroom, can challenge adolescents' autonomy and social status. These interventions may be perceived as attempts to control a personal domain and may trigger adolescents' tendency to assert their autonomy and resist external influences (Henriksen et al. 2006). Developmental research supports the finding that older adolescents, around the age of 16, increasingly assert their right to make personal choices and resist adult control in the school setting, in contrast to younger children aged 8–10. As the students who reported on the school environment were mostly adolescents, this may explain the lack of significant improvement in post-workshop victimization or perpetration scores. However, a significant improvement was observed for bystander intervention.

This suggests that peer mentors, acting as role models, influenced the school environment, leading to increased peer and bystander intervention after the program was implemented. Bystanders and peers play a crucial role in the dynamics of bullying, as their intervention can either stop or perpetuate such behavior (Polanin et al. 2012). The significant improvement in bystander intervention demonstrates the success of the peer-training program in increasing the influence of positive role models among students, fostering a supportive environment, and improving bystander attitudes against bullying.

The qualitative analysis of the open-ended feedback questions suggests that the intervention program was accurately perceived, with students reporting changes in their own attitudes and those of their peers. While many students appreciated the efforts, there were also some students who offered criticism and suggestions. These valuable inputs have been taken into account and have led to the development of intensive plans to tackle cyberbullying, implement a zero-tolerance antibullying policy, improve counselling services, and improve communication through professional development training for teachers. In summary, the qualitative findings demonstrate the potential of peer training to improve the overall school climate and reduce antisocial bullying behavior.

## 7. Recommendations

Peer mentors provided positive feedback on several aspects of the workshop, including the content, activities, and quality of instruction. Based on this positive feedback, the researchers determined that the workshop was effectively planned and executed, and it can be utilized in its current form by other groups. However, the workshop's 4 h duration with only 20 min of break time was not well received. Given that the participants were in Grades 3–9, it was challenging for them to sit for extended periods to attend the session. Research has shown that young children have shorter attention spans and require more frequent breaks during educational classes (Brain Balance 2023). Additionally, some students were unable to participate in the discussion session. To address these concerns, future workshops should be conducted in multiple sessions with smaller groups of students, following the standard class ratio of 20 students. The workshop duration should be adjusted to include longer breaks, and consideration should be given to the optimal duration for similar training sessions for students. Due to the lack of a standardized lesson length, which varies considerably depending on age and subject, a consistent session should be delivered in five separate sessions of 40 min each in future training workshops.

In addition, the researchers observed that, even though the entire training for rural schools was conducted in Urdu, the participation of students in discussions was low, and teachers often had to translate the activities into their local regional language (Pushto, the regional language of Khyber Pakhtoonkhwa). As noted by Mahboob (2007), English is Pakistan's third language of communication, while Urdu and local regional languages are its second and first languages, respectively. To facilitate training in rural areas, the workshop was conducted in Urdu, and the pre-post questionnaires were translated into Urdu as well. However, the researchers concluded that local and regional languages should be considered when conducting such training in rural areas. If needed, the trainer should be accompanied by an additional person who can translate the content into the local language. According to Hinduja et al. (2023), although religious teachings, such as those of Islam, promote equality between men and women, social and cultural influences have led to gender inequality and the prevalence of a strongly patriarchal society. A similar observation was made in the current study, where fewer girls than boys participated in peer training. In order to address this issue, it is recommended that events and training be organized in line with the principles of women's rights as derived from the Qur'an and Sunnah. In addition, the researchers found that financial constraints are a significant barrier to girls' participation in educational programs, as male family members are often given preference when finances are limited (Hinduja et al. 2023). To overcome these challenges, it is suggested that the state carry the costs for the training so that girls can also take part more easily. Another possible solution is to provide micro-financing opportunities or monetary

benefits to female students upon completion of their education to improve the current situation and increase access to education for girls (Mujahid et al. 2015).

Johander et al. (2023) emphasized the importance of long-term interventions with continuous follow-up for improved effectiveness, as opposed to short-term interventions. The lack of significant results observed in the present study may be due to the limited duration of the intervention program. Data collection from students took place shortly after the end of the intervention implementation phase, which may not have allowed a comprehensive and accurate assessment of the impact of the program. In order to obtain more accurate and comprehensive results, it is recommended that a follow-up study be conducted, with data collected several months after the intervention, in order to assess the long-term outcomes of the program.

Furthermore, in the current study, peer mentors implemented strategies based on their own understanding, in collaboration with trained teachers, without being required to report accurately on the implementation of the program. To address these concerns, it is recommended that researchers or trainers visit the institutions after the training to ensure that the intervention is being implemented correctly (intervention fidelity). In addition, it is recommended that peer mentors document their activities and engage in self-reflection on the successful implementation of strategies. They should also report these findings to the researchers to allow for feedback and to ensure that the intervention strategies are implemented effectively.

## 8. Limitations

The workshop's trainer and researcher was well-versed in national and certain regional languages but lacked sufficient fluency in Pushto, which was the most commonly spoken language among participants from a rural background. This limitation was noted during the pilot, and it is recommended that an additional trainer/translator be included in future workshops to facilitate instruction in languages where proficiency may be lacking. Another limitation of the study was the small sample size for pre-post comparison tests such as the split plot ANOVA. While there is no recommended minimum sample size, the results are generally considered to be more reliable when the sample size is larger and participants are evenly represented in the different groups. In this study, the number of girls who participated in the program was smaller than the number of boys, which may be considered a limitation. Therefore, it is recommended that future studies use larger samples with an even distribution of participants among the different groups to improve the comparability of the results. In order to fully assess the results of the study, data were collected from students. However, it is advisable to broaden the scope of the evaluation by collecting data from additional stakeholders such as teachers, staff, and others involved in educational institutions. This recommendation aims to ensure more accurate and unbiased evaluations that take into account different perspectives and insights. By involving multiple stakeholders in the data-collection process, a more comprehensive understanding of the study results can be achieved.

## 9. Conclusions

In conclusion, peer training in antibullying interventions can be a powerful tool in creating a safer and more inclusive school environment. By empowering students to lead the charge against bullying, educators can help to create a culture of kindness and respect that will benefit all students. In addition, peer training can help students to develop important leadership skills, such as communication, conflict resolution, and teamwork. These skills can be useful in many areas of life, beyond just the prevention of bullying. It is imperative to note that the implementation and effectiveness of antibullying interventions, including peer training, may vary depending on the cultural and educational context in Pakistan. However, peer training has the potential to be a powerful tool in the prevention of bullying and the promotion of a positive school culture. Based on the findings of this study, it can be concluded that the Sohanjana Peer-Training Module is a promising intervention for

future bullying interventions. Despite being the first peer-training module implemented in Pakistan, by its results, it can be considered effective. By addressing the identified limitations and implementing the recommended suggestions, it is anticipated that future implementations will yield even more robust results.

**Author Contributions:** Conceptualization, S.S. and A.S.-K.; methodology, S.S.; software, S.S.; validation, A.S.-K.; formal analysis, S.S.; investigation, S.S.; resources, A.S.-K.; data curation, S.S.; writing—original draft preparation, S.S.; writing—review and editing, A.S.-K.; visualization, S.S.; supervision, A.S.-K.; project administration, S.S.; funding acquisition, A.S.-K. All authors have read and agreed to the published version of the manuscript.

**Funding:** This research received no external funding.

**Institutional Review Board Statement:** Researchers followed the basic ethical principles and code of ethics of the APA. Ethical review and approval were not required for the study on human participants according to local legislation and institutional requirements. The entire project was reviewed by the second author's research team and no potential harms or conflicts of interest were identified.

**Informed Consent Statement:** Informed consent was obtained from all subjects involved in the study. School authorities and parents were informed of the purpose of the study, anonymity, free participation, intended use of data, and the right to terminate participation without negative consequences by sending an informed consent form. Parents explicitly agreed to these study terms by signing and returning the form. Written informed consent has been obtained from the parents and school authorities to publish this paper. This ensured informed consent in accordance with ethical guidelines and federal legislation.

**Data Availability Statement:** The data presented in this study are available on request from the corresponding author. The data are not publicly available due to restrictions.

**Acknowledgments:** The researchers acknowledge the support from the German Research Foundation and the Open Access Publication Fund of TU Berlin.

**Conflicts of Interest:** The authors declare no conflict of interest.

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
