# Peer review of "The Sohanjana Antibullying Intervention: Pilot Results of a Peer-Training Module in Pakistan"

_socsci, doi:10.3390/socsci12070409_

Round 1

Reviewer 1 Report

The paper is well-written and appropriately reports the findings of the use of the training program.

My concern is not with the mechanics of the study or the writing, but with the application of a test of the program as a focus for the study rather than changes in pro-social by-stander intervention activities and follow-up measures of the by-stander's self-perception of successful intervention, the bullied-person's self-report of satisfaction with the by-stander's intervention, and, if possible, self-reports of the bullys' satisfaction with the intervention (difficult to impossible to do, but some form of qualitative data collection and analysis might be doable)

Reviewer 2 Report

Dear Authors,

Thank you for the opportunity to review this important manuscript on a complex and important issue as bullying is. 

I think that you could add the informations about the sample (es., gender) also in the Sample section, to make it clear to the readers, even if you add this information later. The same is required for other informations you had taken.

Finally, I think that this kind of work have to be implemented with a double view, on students and teachers. 

Reviewer 3 Report

Review of "The Sohanjana Antibullying Intervention: Pilot Results of a Peer Training Module in Pakistan" = Overall, the article has some strengths in addressing the importance of peer training and bystander interventions in the context of bullying in Pakistani schools. However, there are several areas for improvement, including clarifying the research objectives, organizing the information more coherently, providing specific details about interventions and outcomes, and incorporating relevant citations to support the statements made.

Abstract: It lacks specific details about the findings and outcomes of the pilot study.

Introduction: The introduction provides a background on bullying, emphasizing its impact on students and the need for comprehensive interventions. It mentions the importance of school climate and the role of bystanders in bullying incidents. The introduction highlights the lack of resources in low to middle-income countries like Pakistan for implementing school-wide interventions and suggests peer training as a cost-effective alternative.

Theoretical Background: The theoretical background section provides a comprehensive overview of bystander intervention models and the roles of bystanders in bullying situations. It explains the bystander effect and the sequential steps involved in bystander intervention. The section also discusses the reluctance of bystanders to act and report incidents, particularly in the context of Pakistan. 

Peer-Supported Interventions in Pakistan: This section highlights previous research on bullying in Pakistani educational institutions and the need for peer-supported interventions. It mentions the reluctance of victims to share their problems with adults and the importance of peer support. The section also discusses the success of peer involvement in antibullying interventions and emphasizes the need for training high-status peers. However, the section could provide more specific details about existing interventions and their outcomes.

Summary of the Components of the Peer Training Module: This section provides an overview of the components of the peer training module implemented in the Sohanjana Antibullying Intervention. It mentions general awareness-raising techniques, educating students about participant roles and group dynamics, online and hands-on training for safe social media use, role play activities, and the analysis of real-life bullying cases. However, the section lacks specific details about the content and structure of each component.

Research Methodology: Instruments - the section lacks clarity and needs to be rephrased for better understanding. The authors should provide more information about the development process of the open and closed-ended questions. 

Data Analysis and Results: The section should provide more details about the data analysis methods used, such as how the Wilcoxon sum means, Split Plot ANOVA, and descriptive statistics were applied. It would be helpful to explain the rationale for using these specific statistical tests.

Descriptive statistics: The section should include more information about the closed and open-ended feedback results, particularly the specific findings related to strengths and drawbacks of the workshop.

Pre-Post Comparisons: The description of the Wilcoxon sum-of-means test could be clearer and more concise.

Comparison of School Settings: The explanation of the Split plot ANOVA test should be more detailed and provide a clearer understanding of how the analysis was conducted. The findings about the significant variance between urban and rural students need to be elaborated, providing more specific details and supporting evidence.

Gender differences: The description of the split-plot ANOVA test and its results could be more explicit. The discussion of gender differences needs further clarification and more extensive analysis, including the significance of the observed variations.

Discussion: The discussion section should expand on the findings and provide a more comprehensive analysis of the results.

Recommendations: The recommendations section should provide more specific and actionable suggestions for improving the workshop, such as proposing an appropriate duration and discussing alternative strategies to facilitate participation. The mention of Pakistani culture and girls' involvement needs to be discussed further, drawing on relevant research and addressing possible solutions.

Conclusion: It would be beneficial to emphasize the practical implications and significance of the research and suggest future directions for further investigation. Overall, the article would benefit from providing more context, clarifying certain sections, and supporting the statements with additional evidence or references. Additionally, expanding the discussion and recommendations sections would strengthen the article's contribution to the field.

Round 2

Reviewer 1 Report

The authors addressed the reviewers concerns.